# Plant Cell Wall Hydration and Plant Physiology: An Exploration of the Consequences of Direct Effects of Water Deficit on the Plant Cell Wall

**DOI:** 10.3390/plants10071263

**Published:** 2021-06-22

**Authors:** David Stuart Thompson, Azharul Islam

**Affiliations:** School of Life Sciences, University of Westminster, 115 New Cavendish Street, London W1W 6UW, UK; azharul.mywestminster@gmail.com

**Keywords:** plant cell wall composition, expansins, water stress, salt stress

## Abstract

The extensibility of synthetic polymers is routinely modulated by the addition of lower molecular weight spacing molecules known as plasticizers, and there is some evidence that water may have similar effects on plant cell walls. Furthermore, it appears that changes in wall hydration could affect wall behavior to a degree that seems likely to have physiological consequences at water potentials that many plants would experience under field conditions. Osmotica large enough to be excluded from plant cell walls and bacterial cellulose composites with other cell wall polysaccharides were used to alter their water content and to demonstrate that the relationship between water potential and degree of hydration of these materials is affected by their composition. Additionally, it was found that expansins facilitate rehydration of bacterial cellulose and cellulose composites and cause swelling of plant cell wall fragments in suspension and that these responses are also affected by polysaccharide composition. Given these observations, it seems probable that plant environmental responses include measures to regulate cell wall water content or mitigate the consequences of changes in wall hydration and that it may be possible to exploit such mechanisms to improve crop resilience.

## 1. Introduction

It is generally accepted that plant cells expand by slow irreversible deformation of their cell walls as a result of stresses generated in the walls by the internal turgor pressure of the cell [1], and thus their growth can be framed as a biomechanical interaction between the properties of the wall of a cell and its turgor pressure. Water stress is expected to affect plant growth as a result of effects on either wall stress or wall mechanical behavior.

The turgor pressure of a cell is determined by its internal osmotic pressure and the water potential outside the cell. For a non-growing cell, it is simply the sum of these terms; however, in a growing cell there must be an imbalance favoring water movement into the cell, although the difference is likely to be slight at a cellular level. Because water potential influences turgor pressure, it is clear that the water status of a plant can alter wall stress and that this can affect the rate of plant growth; however, it should be noted that in many cases the concentration of cellular solutes increases to maintain turgor pressure [2]. There are also many examples of the active modulation of wall mechanical properties in response to signals associated with water stress or water stress itself (e.g., [3]). However, even though water makes up the majority of the primary walls of plant cells by mass (typically >80%, [4]), the direct effects of the water status of a plant on its wall properties have been relatively unexplored. Given the proportion of primary cell walls of plants that water comprises, it would be surprising if the volume that it occupies does not contribute to the mechanical behavior of the wall.

There is good reason to expect that changes in the water content of plant cell walls could affect their mechanical behavior directly. In polymer matrices, the free volume between polymer molecules is thought to have substantial effects on their properties and behavior. This effect is routinely exploited to control the properties of plastics by the addition of the smaller spacing molecules, termed plasticizers (e.g., [5]). Plant cell walls comprise a composite of polymers, with the space between them generally occupied by water (although water may be displaced by other materials in secondary wall deposition).Therefore, water appears analogous to plasticizers in synthetic plastics. For example, pectin films become several-fold stiffer as their degree of hydration is reduced [6].

Tests of the effect of altering the water content on cell walls of rye coleoptiles [7] and sunflower hypocotyls [8] by using osmotica have shown the rate at which walls extended under constant load was reduced as water was removed from the walls and that these effects occurred at water potentials that plants might be expected to experience under “field” conditions (−0.4 MPa in sunflower). The tissue in both sets of experiments had been frozen and thawed, and so the effects of reduced turgor pressure and of cellular responses to the altered water potential could be excluded and the observed effects attributed solely to direct effects of hydration on the cell walls. Substantial reductions in hydration were found in tomato fruit pericarp cell wall material at even more moderate water potentials, with hydration decreasing by more than 50% as the water potential was reduced from 0 MPa to −0.15 MPa [9].

If these effects occur in vivo, changes in wall hydration, spacing and tissue biomechanics as a result of plant water status would be expected to have a direct impact on plant growth. Additionally, reducing the available space in the wall might impede the movement of enzymes and other molecules within the wall [10] and amplify the water potential gradients across tissues if apoplastic hydraulic conductivity decreases because of reduced wall area (e.g., [11]). It would therefore be surprising if responses to water availability do not include measures to maintain wall hydration or to mitigate the consequences of changes. The exploration of such adaptations may shed light on plant responses to water availability and perhaps also offer ways to maintain crop yields and improve resilience when water supplies are limited.

Such measures might include changes in wall polysaccharide composition, structure or charge to alter the relationship between wall water content and water potential, regulation of the osmotic pressure, pH or ion concentrations in the wall space to maintain hydration, or wall “loosening” to minimize any effects of hydration changes on extensibility.

One potential mechanism for controlling the water content of a cell wall is by altering its composition. In principle, this might involve changes in which polysaccharides are present or their relative proportions, but changes in polymer length and the number and composition of branches may also alter these interactions. For example, the quantity of water bound per monosaccharide unit of chitosan, alginate and cellulose in paper differs, as does the way that this relationship changes with water activity [12], such that they dehydrate at different rates as the water potential is reduced. In practice, the pectins in the wall possess the greatest chemical and structural complexity and thus potentially offer multiple mechanisms for modulating these relationships, as does the degree of esterification of pectic acidic groups.

The primary factors determining this relationship between water potential (or water activity) and the quantity of water associated with polymers are the space occupied by the polymer (itself affected by polymer length, both the number and length of branches and the mobility of polymer segments) and the strength of interactions between the polymer functional groups and water [12]. Interactions with groups at chain ends (and therefore the number of chain ends) may also be important [13]. All of these properties could be modulated by alterations in wall biosynthesis or by enzymes acting upon existing wall material.

In addition to the individual interactions between the polymers and water, the cell wall as a whole has properties resulting from interactions between its polymer components, conferring resistance to compression or conversely reducing chain separation or mobility. It certainly seems that a number of wall components have roles in maintaining the spacing required for correct wall function [14]. This “scaffolding” may be necessary to prevent lateral collapse of walls as they deform, but also potentially to prevent compression of walls by the turgor pressures of adjacent cells pushing against one another in tissues exhibiting tissue pressure/tension [15]. Such effects on hydration as a result of interactions between polymers would be expected to include the “egg box” bridges formed by divalent cations between pectic uronic acid groups [16] as well as factors affecting the strength of these interactions, such as ionic strength and pH, if this alters the density of charged groups [9].

Cell wall enzymes offer more rapid mechanisms for modifying the relationship between water potential and wall water content than could be achieved by bulk changes in wall composition. Many types of modification could have such effects, including (but not limited to) changes in polymer molecular weight, degree of polysaccharide branching and length of branches, the number of polymer ends, and the density of charged groups. Another group of cell wall enzymes of considerable interest are the wall loosening proteins known as expansins, because it has been observed that in addition to making plant cell walls and other cellulosic materials more extensible at pH < 5.5 (e.g., [17]), expansins also cause swelling in these materials [18,19]. It therefore seems likely that, in principle, expansins could maintain or increase wall spacing under conditions of water stress.

The following experiments explore the effects of water potential on water content and mechanical behavior and of wall composition and expansins, and the interaction between them, upon these relationships in both plant materials and synthetic models of plant cell walls based on bacterial cellulose. These data establish that the water potentials that plants may experience under field conditions can directly affect the mechanical characteristics of their cell walls in ways that would be expected to affect plant growth and development in vivo and that these effects can be modulated by wall composition and expansin activity.

## 2. Results

In classical experiments, the pore size of plant cell walls was determined by observing whether polyethylene glycols (PEGs) of a range of molecular weights caused plasmolysis or cell collapse, called cytorrhysis [20]. Smaller M_w_ PEG molecules penetrated the cell wall and exerted an osmotic pressure at the cell membrane, causing plasmolysis, whereas larger M_w_ PEG molecules were excluded from the wall and caused cytorrhysis. However, osmosis depends on the exclusion of solute, and so these experiments also demonstrated that because PEG molecules with a molecular weight of >4 kDa could not penetrate plant cell walls, they could be used to apply osmotic potentials to them and modify their water content. This method has previously been used to show that plant cell wall material becomes less extensible if PEG is used to reduce its water content in constant-load extensiometer (also known as creep) measurements [7,8,19]. Here, we examine the effects of reduced water potentials generated using PEG on the retention of water and mechanical behavior in bacterial cellulose and composites of bacterial cellulose with pectin and xyloglucan in order to investigate the effects of incorporating these cell wall components.

Figure 1a illustrates examples of the effect of reducing the effective water potential of buffers bathing the bisected frozen and thawed sunflower hypocotyls extending under an applied load. In these experiments, the applied water potential was reduced from 0 MPa to −0.62 MPa or was increased from −0.62 MPa to 0 MPa by exchanging control buffers and buffers containing PEG 6000. Note that the solutes in the buffer should penetrate the wall freely and therefore will not exert an osmotic potential on the wall. Reducing wall water potential, and therefore water content, caused a transient increase in length, followed by a substantial reduction in the long-term rate of extension most analogous to growth in its rate and the period over which it occurs. Increasing the water potential caused a rapid increase in the long-term extension rate after a short lag. It was reasonably common to see a slight shortening of the cell wall material during this lag period (illustrated in detail in Figure 1b, along with the initial increase in length when water content was reduced), and so it seems likely that it mirrors the lengthening during reductions in water content (although of lesser magnitude) and that these effects result from lateral realignment of wall components as the total volume in which they are contained changes.

Sample data from similar experiments using bacterial cellulose and bacterial cellulose composite with apple pectin and tamarind xyloglucan (CPX) are shown in Figure 1c,d respectively. In these experiments the buffers were exchanged twice so that either the control buffer was replaced with buffer containing PEG for a period before returning the strips to the control buffer or the buffer containing PEG was replaced for a period with the control buffer before returning the strips to the buffer containing PEG. As for the sunflower material, the PEG generated an osmotic potential of −0.62 MPa.

Although the bacterial cellulose was much less extensible than the sunflower walls, and the changes in rate of extension are therefore less clear than in the plant cell wall material, the same broad effect of water content was observed.

In CPX, long-term extension was substantially reduced when the water content of the material decreased, as was seen in both the plant cell wall material and the bacterial cellulose. However, the material was considerably more extensible than cellulose without pectin and xyloglucan (note that these data are for a lower load than those for the sunflower walls and bacterial cellulose) and exhibited substantial rapid extension after the initial water potential changes, both when the osmotic potential was decreased and when it was increased. The increase in length that occurred after strip hydration was decreased parallels that seen in plant material and in bacterial cellulose alone, although it is considerably enhanced compared to bacterial cellulose. In other experiments, strips of bacterial cellulose composites, which included xyloglucan or other hemicelluloses (including lichenan and mixed-linkage β-glucan), consistently narrowed to a much greater extent than strips of bacterial cellulose or cellulose composite with pectins during constant load extensiometry, suggesting that these hemicelluloses reduced the resistance of the material to lateral compression (in engineering terms, an increase in their Poisson’s ratio, which is derived from the ratio of lateral contraction to longitudinal extension as material is stretched (Poisson’s ratio for extension in two directions = −dε_lateral_/dε_longitudinal_, where dε_longitudinal_ is longitudinal strain and dε_lateral_ is lateral strain; dε_lateral_ is negative because it is a contraction, and so a negative sign is included to give a positive value for the Poisson’s ratio).

The extension when the strip water content was increased appears to be a different process (or processes) and can be modelled extremely well (r^2^ > 0.99) by assuming processes in which an initial extension rate decreases exponentially, reducing by a factor of *e* over a period known as a retardation time (in engineering terms, Kelvin or Kelvin-Voigt elements (a Kelvin element extends with a rate at time t = r_0_(*e*^−t/τ^), where r_0_ is the extension rate of the element at time = 0 and τ = retardation time. Length at time t = r_0_ × τ(1 − *e*^−t/τ^), which is obtained by integrating the previous equation with time). In this case, the model employed two Kelvin elements with retardation times of approximately 5 min and approximately 64 min. These are comparable to those found for creep of plant cell wall material extending under a constant stress [21].

In such rheological models, the individual mathematical elements have often been interpreted as corresponding to different processes by which the material extends (such as the stretching of polymers and realignment of polymers). Therefore, the correspondences with the model of creep in plant material suggests that this “released” extension may indicate equivalent mechanisms of extension. If so, it appears that they were enhanced by incorporation of other wall polysaccharides into the bacterial cellulose matrix. The flow rates of both Kelvin elements in cell walls from tomato fruit epidermis increased at lower pH and were reduced by boiling, indicating that they may have been enhanced by expansins [21]; the pattern of extension after the first increase in water potential also resembles that previously reported in composites of bacterial cellulose treated with expansin [17].

The loss of water from pieces of bacterial cellulose, bacterial composite with apple pectin (CP), and CPX at reduced water potentials was then determined by measuring their change in weight with time after immersion in buffer containing PEG to give a water potential of −0.62 MPa. These data are illustrated in Figure 2a,b, as is their recovery after the pieces were returned to the control buffer without PEG. The proportional changes in weight are summarized in Figure 3. It was consistently observed that none of the materials recovered their original weight and water content completely after they were returned to the buffer that did not contain PEG, exhibiting some degree of hysteresis until they were treated with expansin (the cucumber α-expansin CsExp1 in Figure 2a and Figure 3a) or snail powder extract (in Figure 2b and Figure 3b). Snail acetone powder from the visceral hump of *Helix pomatia* has previously been reported to possess an expansin-like activity and to contain proteins labelled by polyclonal antibodies to purified cucumber expansins [22]; thus, snail powder extracts were used as a second source of expansin activity.

Such hysteresis has been seen in other studies of plant cell wall dehydration and rehydration [4]. From these data it seems that space is lost during water removal and reswelling, either because of additional interactions between the cellulose and other components in the composites or because the forces driving reswelling are insufficient to fully restore the original volume, or both, but that these factors can be negated by expansins.

It seems unlikely that that the differences between the materials were because of variations in exclusion of the PEG (at least within the period of these experiments) because the effect of a large but permeating osmoticum (PEG 1000) on sunflower cell walls was initial water loss followed by partial recovery as the osmoticum diffused into the wall (see Appendix A, Figure A1). This was not observed for these cellulose, CP, or CPX batches, where recovery did not occur until after the materials were returned to the control buffer (although it was seen in batches of bacterial cellulose or cellulose composite produced in incubations of less than 120 h).

Therefore, it seems that incorporation of xyloglucan substantially enhanced the effect of lower water potentials on water content and the magnitude of these changes; this may be reflected in the comparatively large rapid changes in length when water content of CPX strips was increased and decreased for the first time (Figure 1d). There were variations between batches of the cellulose and composites, and the CP batch used for the data in Figure 2b and Figure 3b retained water better than cellulose alone.

In other experiments, the composite produced using polygalacturonic acid (Sigma-Aldrich, cat. No. 81325) lost considerably less water than the composite with apple pectin in PEG solutions, giving an osmotic potential of −0.62 MPa for 180 min (composite with apple pectin lost 34.11 ± 2.26% and composite with polygalacturonic acid lost 20.16 ± 2.61%). This indication that charge density can affect these interactions raises the question of whether apoplastic pH could also affect the wall volume by altering the proportion of pectin uronic acid groups that possess a negative charge (this has been suggested by observation of the effect of pH on hydration of tomato cell wall material [13]). Observations of the effect of pH on the viscosity of apple pectin solutions/gels in the presence of Ca^2+^ from flow-through tests (Figure 4) and of its fluidity in inverted tubes (Appendix A, Figure A2) appear to confirm that the properties of pectins are altered as pH is increased within potentially physiological wall conditions between pH 4 and pH 6, and it seems probable that this is because of Ca^2+^ cross-linking uronic acid groups.

It is possible to observe the effect of expansin on the volume of plant cell walls from an increase in the optical density (O.D.) of suspensions of fragments of cell wall material in buffer. This is illustrated for fragments of wall material prepared from etiolated sunflower hypocotyls in response to 14 µg of CsExp1 and 200 µL of snail powder extract in Figure 5a. The increase is extremely rapid and is largely complete within 2 min of adding expansin. The mean change in optical density 2 min after treatment with CsExp1, snail powder extract horseradish peroxidase (2 units), and hydrogen peroxide (final concentration = 84 mM H_2_O_2_) are shown in Figure 5b. Note that the cell wall material was boiled to inactivate endogenous expansins before homogenization. The effects of peroxidase and hydrogen peroxide, both of which caused slight reductions in O.D., are included for interest, because it has been reported that peroxidative cross-linking of cell wall components such as extensins causes a reduction in cell wall volume [23].

To determine whether cell wall composition could affect this response, suspensions prepared from leaves and petioles of Col-0, *mur1-2*, and *mur2* Arabidopsis plants were treated with CsExp1. Both the *mur-1* and *mur-2* mutants have cell walls with reduced fucose content. MUR-1 encodes the GDP-D-mannose-4,6-dehydratase required for synthesis of GDP-L-fucose [24] and MUR-2 for the fucosyltransferase 1 that fucosylates xyloglucan in Arabidopsis [25]. Therefore, Arabidopsis *mur1* mutants haven substantially reduced fucose content in xyloglucan, rhamnogalacturonan I, rhamnogalacturonan II (RGII), and arabinogalactan proteins, but in *mur2* mutants the reduction is restricted to xyloglucans. Although the leaf cell walls of *mur1* mutants contain normal amounts of RGII, only half forms the borate cross-linked dimer, whereas almost 95% of the RGII forms the borate dimer in wild-type plants [26]. The *mur1* phenotype can be rescued by treatment with elevated levels of boron [26,27].

Under the growth conditions used, the Col-0 and *mur2* plants appeared indistinguishable, but the *mur1-2* plants were noticeably smaller. It was also observed that cell wall fragments from *mur1-2* plants exhibited less swelling in response to CsExp1 than those from Col-0 and *mur2* plants (Figure 6a).

No differences were observed between the effect of expansin on cell wall fragments from Col-0 and *mur2*, and pretreating *mur1-2* cell wall fragments with boric acid partially restored swelling in response to expansin (Figure 6b), strongly suggesting that at least part of the reduction in expansin-mediated wall swelling in material from *mur1-2* seedlings was because of its effect on the side chains of RGII and their capacity to form borate cross-linked dimers [26,27,28,29,30]. These data also establish that, in principle, expansin-mediated swelling is influenced by wall composition.

## 3. Discussion

In addition to the well-established impacts of water stress on plant growth as a result of reduced cell turgor pressure and active environmental responses, it is likely that there are also previously unexplored effects of water deficits on cell wall polymer spacing and wall mechanical behavior. Indeed, the function of some observed stress responses may be in part to stabilize wall spacing or to compensate for changes.

As was noted in the introduction, such responses might include changes in wall composition as a result of de novo wall deposition or modification of existing wall components as well as the action of expansins. Additionally, while pH might have a range of effects, it notably affects expansin activity [31]. Ion concentrations could both cause osmotic effects, and in the case of Ca^2+^ ions, mediate cross-linking between pectin uronic acid groups [9].

The results presented demonstrate a profound effect on wall water hydration at reduced water potentials as a result of incorporation of xyloglucan into bacterial cellulose composites; thus, polysaccharide composition can clearly determine relationships between water potential and spacing in materials analogous to plant cell walls. The substantial short-term changes in extension and changes in hydration in CPX compared to bacterial cellulose alone hint that (at least in these materials) xyloglucan, and perhaps other hemicelluloses, facilitate realignment of the cellulose under applied forces, whether extensive (from the extensiometric data) or compressive (from the effects of water potential on water content).

However, as a result, incorporating xyloglucan into bacterial cellulose composites substantially increased the effect of water potential on their degree of hydration, and so (unless their effects are markedly different in plant cell walls) it seems likely that they (and perhaps other hemicelluloses) are more likely to accentuate the effects of water potential on spacing than to mitigate them (although if they facilitate realignment of wall components, this may mitigate the mechanical effects of reduced spacing).

Perhaps the most obvious candidates for the role of maintaining hydration are the pectins. Pectins are amongst the most structurally complex of the cell wall polysaccharides and additionally carry varying charge density and distribution, depending on the degree and distribution of esterification of the uronic acid groups. Therefore, pectic interactions with one another and other wall components, variation in their size, degree, and nature of branching, and the effects of ion concentrations and pH on them offer multiple mechanisms for modulating hydration of the wall. It has indeed been found that specific modification of rhamogalacturonan I (RGI) structure by introduction of a β-1,4-galactanase into potato tubers increased the rate of water loss compared to controls and made the material more brittle in mechanical compression tests [32]. We found that cellulose composites with polygalacturonic acid that has been de-esterified retained a greater degree of hydration at an osmotic potential of −0.62 MPa than composites with apple pectin, suggesting that cross-linking can also strengthen the material against collapse. These experiments were in the presence of Ca^2+^ ions, but incorporation of pectins into bacterial cellulose composites increased their resistance to compression even without Ca^2+^, though they became less stiff to shear oscillation [33].

Charged uronic acid groups could affect the relationship between volume and water potential in a number of ways, including charge repulsion, cross-linking via divalent metal ions [34,35], and generating osmotic pressures from cations immobilized by charge interactions (a phenomenon known as the Donnan effect [35]). Hydration of tomato fruit cell wall material was found to decrease with increasing concentration of NaCl (with the greatest change between 0 and 2.5 mM) in 6 mM CaCl_2_ and in buffers with pH > 4.5 [9]. These observations suggest a complex relationship between pectic charge and volume affected by Donnan effects at low ionic concentration as well as cross-linking by divalent ions (most importantly Ca^2+^ in vivo). Increasing pH led to a reduction in wall swelling, within ranges that might be expected in vivo (between pH 4 and pH 5). This is higher than might have been expected from the pK_a_ of polygalacturonic (which is typically reported as <3.5; [9]) but is in accord with the data presented in Figure 4 and Figure A2. Ca^2+^ cross-links form cooperatively [36], and so effective cross-linking may depend on the probability that there are clusters of adjacent de-esterified uronic groups, “shifting” the pH at which cross-linking is affected into physiological ranges.

### 3.1. Water Stress and Cell Wall Composition

If there is a direct effect of plant water potential on wall properties, it seems to be worth examining examples of reported changes in cell wall composition, gene expression, and enzyme quantity or activity in response to water stress for possible evidence of regulation of wall hydration and spacing.

Changing wall composition in response to the reduction of water availability has been reported in a number of instances. Several-fold increases in labelling of xyloglucan and unesterified pectin have been reported during dehydration and rehydration in the resurrection plant *Craterostigma wilmsii* [37], and an increase in arabinose-rich pectins and arabinogalactan proteins have been found in a range of other resurrection plant species [38]. Less extreme examples include an increase in pectic branching in response to a water stress of −0.5 MPa in walls of root tips of a drought tolerant cultivar of wheat, which was not seen in a susceptible cultivar [39], and changes in the polysaccharide composition and molecular weight distribution in cell walls of wheat coleoptiles in response to PEG-induced water stress [40] (although the osmotic potential is not given, we estimate that it would be approximately −0.5 MPa for the concentration of PEG specified). Of these, increasing pectin branching is certainly one type of response that would be expected to maintain wall hydration and spacing.

Researchers have also reported a range of changes in the activity of enzymes capable of acting upon plant cell walls in response to water or osmotic stresses, and in many cases these enzymes would be expected to be involved in remodeling the walls. For example, it seems that xyloglucan transglycosylase-hydrolases (XTHs) and xyloglucan endotransglycosylases (XETs), enzymes involved in modifying xyloglucan chain length, often increase under these conditions. XET activities were found to increase in maize roots close to the tip, where growth was maintained at extreme water deficit conditions (−1.6 MPa), and both growth maintenance and increased XET activity required ABA [41]. Similar increases in XET have been observed in elongating regions of roots of wheat seedlings [42] and in leaves of *Lolium temulentum* L. seedlings [3] experiencing water deficit. Evidence that XTHs are involved in the adaptation of plant cell walls to conditions of water deficit comes from the observation that expression of an XTH gene induced by a range of abiotic stresses in hot pepper conferred resistance to drought and salt stresses in tomato [43].

Whether such changes in XTH or XET activity will indeed affect wall water retention under water stress is, at present, speculative and would be expected to depend on their exact effects on chain length and number of ends. However, the data in Figure 1 and Figure 2 suggest that xyloglucan can have a profound effect on these relationships.

### 3.2. Expansins and Wall Spacing

It is clear that expansins can increase wall spacing, and so the many reported cases of expansins increasing under drought stress are striking. For example, changes in expression in a range of expansins were found in potato in response to ABA and drought (as well as a number of other plant hormones and abiotic stresses), with some genes being upregulated and some downregulated [44]. Expansin activity and transcript levels of a number of expansin genes were also observed to increase in response to a reduced water potential (−1.6 MPa) in the regions of maize root tips where growth continued at these considerable water stresses [45,46]. ABA and osmotic stress generated using PEG increased expression of one expansin (EXP1) in elongating tissues of soybean during germination [47] and both expansin activity and expression of one expansin gene (TaEXPA1) under moderate osmotic stress in wheat leaves, an effect that was markedly greater in a drought-resistant cultivar than one that is more susceptible to drought [48].

These changes could be to increase wall extensibility and thereby compensate for a reduction in cell wall stress due to decreased turgor, but in many cases plant cells increase cell osmotic pressure under conditions of reduced water availability so that turgor pressure and wall stress are substantially unaffected [2,49,50] (although this is not always the case, e.g., [51]). Another hypothetical possibility might be that expansin quantities increase to offset a reduction in activity caused by the apoplastic alkalinization often associated with water stress [52]. However, the data in Figure 2, Figure 3, Figure 5 and Figure 6 show that expansins can increase and restore cell wall spacing and that the magnitude of this effect depends on wall composition. If this is the case, and reduced water potentials affect cell wall spacing, it appears at least plausible that the functions of expansins under conditions of water stress include maintenance or restoration of wall space or mitigation of the effects of spatial constraint on wall stiffening, or both.

Spacing between cell wall components may also be lost as they realign during wall deformation, and given the observed effect of expansins on hysteresis during dehydration and rehydration cycles, it is conceivable that they play a role in controlling the strain hardening that this could cause.

### 3.3. RGII

RGII comprises a very small proportion of plant cell walls [53], but the *mur1* Arabidopsis mutants, in which its structure is altered, have a drastically dwarfed phenotype. The *mur1* phenotype can be substantially rescued by supplying boric acid; it seems that this is because of a reduction in the borate diester cross-links formed between RGII domains via apiose residues [27].

In the experiments reported here, we found that the swelling of cell wall fragments from *mur1-2* in response to expansin was inhibited compared to wall fragments from wild-type plants. Partial restoration of the response by pre-treatment with boric acid strongly suggests that the difference between wild-type and *mur1* material was because of altered RGII cross-linking. However, cell wall swelling and increased porosity have been reported to result from boron deficiency [54,55].

This apparent contradiction could be because these experiments tested the change in swelling of wall fragments rather than their absolute size, and so the reduced effect of expansin could be because they were already more swollen than those from the control seedlings. However, the observation that salt stress causes root cells to burst in *mur1* [56] and some other Arabidopsis mutants with altered RGII structure [57] suggests the possibility that under some conditions wall swelling might lead to excessive plasticization of the walls. Na^+^ ions might cause such an effect if they become immobilized by interaction with uronic acid groups and generate an osmotic pressure in the wall (a Donnan effect) or by competing with Ca^2+^ ions for uronic acid groups, thereby reducing cross-linking between pectins other than RGII [58].

Partial restoration of the expansin response by pre-treatment with boric acid strongly suggests that the difference between the effect of expansin on wild-type (and *mur2*) and *mur1* material was because of altered RGII cross-linking. This may have been because the fragment size was reduced by the pre-treatment, giving a lower baseline so that expansion to the same final fragment size gave a greater proportional effect (it is hard to be sure of this because some O.D. was always lost during the washing steps used to remove the boric acid), but it may be worth considering whether RGII cross-linking also contributes to the force required to drive cell expansin-mediated wall swelling and whether RGII cross-linking may mediate a balance between compressive and expansive forces in plant cell walls.

### 3.4. Summary

It appears that plant cell walls are plasticized by water and that water potentials experienced under field conditions can alter wall water content to a degree that could affect plant growth and development via this mechanism. If this is the case, then it is to be expected that adaptations to conditions of low water availability or environmental fluctuations in water availability will include measures to maintain wall hydration or mitigate the effects of water potential on wall spacing.

We believe that this model offers a novel perspective for interpreting the role of the cell wall in plant adaptations to water stresses. A better understanding of such relationships between plant cell wall composition, hydration, and biomechanics may shed light on a relatively unexplored aspect of plant responses to their environment and offer new targets for improving crop resilience in the Twenty-first century.

## 4. Materials and Methods

### 4.1. Plant Material

Sunflower seeds (Giant Yellow variety, Suttons seeds, Paignton, UK) were imbibed in tap water overnight and then sown in pots containing water-saturated perlite (Silvaperl, Gainsborough, UK). The plants were grown for 5 days at about 30 °C in pots covered with a plastic pot lid and wrapped with aluminum foil to obtain etiolated hypocotyls.

*Arabidopsis thaliana* wild type (Col-0), *mur-1-2* (N6244), and *mur-2* (N8565) seeds (all from The Nottingham Arabidopsis Stock Centre, University of Nottingham, Nottingham, UK) were surface sterilized in 70% ethanol for 2 min and 5% sodium hypochlorite for 10 min, rinsed five times in distilled water, and then spread onto plates of half strength Murashige and Skoog medium supplemented with 1% sucrose and 0.7% agar. The plates were kept at 4 °C for 4 days in complete darkness for stratification and then grown for about 14 days under a 16 h light/8 h dark photoperiod and 23 °C/19 °C temperature regime.

### 4.2. Bacterial Cellulose and Bacterial Cellulose Composites

Bacterial cellulose was produced by culturing *Komagataeibacter sucrofermentans* (ATCC^®^ 700178™, Promochem, London, UK) in 100 mL YGC medium (50 g L^−1^ glucose, 5.0 g L^−1^ yeast extract, 12.5 g L^−1^, CaCO_3_) in 250 mL Erlenmeyer flasks in static conditions for at least 120 h at 26 °C. A long incubation was necessary to ensure that the pellicles were dense enough to exclude PEG of Mw 6000 (PEG 6000). Composites of bacterial cellulose with pectin were produced by the addition of 0.5% (*w*/*v*) pectin to the YGC medium, and composites of bacterial cellulose with pectin and xyloglucan by addition of 0.25% (*w*/*v*) pectin and 0.25% (*w*/*v*) xyloglucan to the YGC medium (apple pectin, Sigma-Aldrich, Dorset, UK, cat. No. 76282, 70–75% esterified; tamarind seed xyloglucan, Megazyme International, Ireland, Cat. No. P-XYGLN).

### 4.3. Expansin and Snail Powder Extract

Expansin was a gift from Professor Simon McQueen-Mason (University of York, York, UK); it was the α-expansin CsExp1 expressed in tomato plants and dissolved in pH 4.5 35 mM MES/35 mM Acetate buffer containing 200 mM NaCl [59].

Snail powder extract was prepared by mixing 50 mg of snail acetone power from the visceral hump of *Helix pomatia* (Sigma-Aldrich, Poole, Dorset, UK, Cat. No. S9764, now discontinued) into 1 mL of MES buffer 10 mM MES buffer containing 5 mM KCl and 1 mM CaCl_2_ titrated to pH 5.0 with 1 M NaOH using a laboratory vortexer and incubating for 10 min at room temperature before centrifuging for 10 min at 7200× *g* and collecting the supernatant.

### 4.4. Extensiometry

Segments of 20 mm were cut from the top growing part of the sunflower hypocotyls, longitudinally bisected using a double-edged razor blade, and then frozen using freezing spray (R.A. Lamb, Eastbourne, UK). After 60 s, the segments were thawed in MES buffer. The segments were then pressed between microscope slides wrapped with absorbent paper, using about 2 kg of weight for 60 s, and returned to MES buffer. Segments of cellulose or cellulose composite 2 mm × 1 mm in cross-section and about 10 mm in length were cut from pellicles using a pair of razor blades held in a spaced block and transferred to MES buffer.

Constant load extensiometry was carried out as described in [21]. Segments of plant material, bacterial cellulose, or cellulose composite were fixed into a reservoir tube with about 10 mm exposed between a clamp at the bottom of the tune and another clamp attached to one arm of a counterweighted cantilever. The reservoir tube was then filled with buffer. The exact initial length of the exposed portion of the segment was measured using a magnifying eyepiece with a graticule (extensiometer custom built by the Biological Sciences workshop at the Lancaster University, Lancaster, UK).

The counterweight positions were adjusted so that a negligible force was applied to the samples for 20 min to allow them to settle, after which the force was increased in increments of 0.098 N (10 g) by sliding the counterweights along the cantilever arms. The change in length was measured using a linear variable displacement transducer (Schlumberger DFG 2.5 from RS Components Ltd., Corby, Northants, UK), which gave a voltage output based on the position of a core balanced on the opposite end of the cantilever arm in relation to the one to which the clamped segment was attached. The mean of 1000 individual readings was recorded every 30 s by a Bede PCADH24 analogue input card (Bede Technology, Jarrow, Tyne and Wear, UK). The extensiometer can be used to test up to six samples at a time. All extensiometry experiments were conducted at least twice, using at least two replicates of each treatment in each run. The data presented are examples of single tests, because although the replicate data exhibited the same patterns of response, they are not normally closely enough matched for pooling of multiple runs to be feasible.

The buffer used throughout (referred to as MES buffer) was 10 mM MES buffer containing 5 mM KCl and 1 mM CaCl_2_, titrated to pH 5.0 with 1 M NaOH. The pH of buffers containing PEG or other osmotica was adjusted to pH 5.0 after their addition. The effect of changing the water potential on segment extension was observed by draining MES buffer from the extensiometer reservoirs and replacing it with the same buffer containing PEG 6000 or by carrying out the opposite exchange. PEG osmotic pressure (and therefore water potential) was determined by vapor pressure osmometry using a Vapor^®^ 5520 osmometer (Wescor Inc., South Logan, UT, USA). Melting point depression measurements of the osmotic pressures of PEG solutions are higher than those made using vapor pressure [60], but it has been plausibly argued that this is because of the effect of temperature on higher order virial coefficients of the Van’t Hoff equation in PEG solutions and therefore that measurements from melting point, which are necessarily at low temperatures, do not accurately reflect the osmotic pressure under the measurement conditions [61], and vapor pressure measurements were preferred. From these data, 0.27 g PEG 6000 g water^−1^ gave an osmotic potential of −0.62 MPa.

### 4.5. Viscosity of Pectin Solutions

The viscosity of 2% (*w*/*v*) solutions of apple pectin (Sigma-Aldrich, cat. No. 76282, 70–75% esterified) containing 5 mM CaCl_2_ adjusted to the required pH by dropwise addition of 2 M KOH were tested by timing how long the solutions took to flow between two graduations of a 1 mL graduated glass pipette. The same pipette was used throughout for consistency and was wetted with the pectin solutions before measurements were carried out. The viscosity of 2% (*w*/*v*) solutions of apple pectin (Sigma-Aldrich, cat. No. 76282, 70–75% esterified), containing 10 mM CaCl_2_ adjusted to the required pH by dropwise addition of 2 M KOH, were also visualized by inverting capped disposable centrifuge tubes for 5 min. Measurements were carried out at room temperature.

### 4.6. Measurement of the Effects of Water Potential on Water Content of Bacterial Cellulose and Cellulose Composites

Segments of bacterial cellulose or cellulose composite, 2 mm × 1 mm in cross-section and approximately 10 mm in length, were cut from the cellulose and cellulose composites with a pair of razor blades held in a spaced block and transferred to MES buffer for at least 10 min at room temperature. Segments were then blotted quickly using absorbent paper to remove surface buffer, and their initial weight measured quickly using a microbalance. The pieces were then transferred into Eppendorf tubes containing 800 μL of MES buffer containing PEG 6000 solution (0.27 g PEG 6000 g water^−1^, giving an osmotic potential of −0.62 MPa, determined using a Vapor^®^ 5520 osmometer as noted above) and the weight changes were measured at regular intervals for 180 min by removing the segments from the buffer, quickly rinsing them in MES buffer without PEG to remove the viscous PEG solutions, blotting them to remove surface liquid, and again weighing them before returning them to the tubes of MES buffer with PEG. After 180 min, the pieces were transferred to Eppendorf tubes containing 800 μL of control MES buffer without PEG to rehydrate, and the weight changes again were recorded at regular intervals. Finally, 11.2 μg of α-expansin or 200 μL of snail powder extract was added to the MES buffer, and any further changes in weight were recorded.

### 4.7. Cell Wall Swelling Measurements from Optical Density

Cell wall fragments suspended in buffer were prepared from approximately 1 g of segments from the top 2 cm of sunflower hypocotyls or approximately 1 g of Arabidopsis seedling leaves and stems. The material was boiled for 90 s in distilled water to inactivate endogenous expansins in the plant tissues and then homogenized in 10 mL of MES buffer using a laboratory mixer/emulsifier at full speed for 5 min (Silverson Machines Ltd., Waterside, UK). The homogenized suspension was then centrifuged for 2.5 min at 650× *g* to remove large pieces of material and then centrifuged again for 10 min at 1450× *g* to pellet smaller cell wall fragments of cell wall. These were re-suspended in 10 mL MES buffer. Fragments from Arabidopsis seedlings were pelleted by centrifugation at 1450× *g* and resuspended in 10 mL MES buffer two further times to eliminate pigmentation in the suspensions (although there was negligible absorbance from them at a wavelength of 750 nm).

The optical density of the fragments suspended in buffer at a wavelength of 750 nm was determined using a spectrophotometer (Lambda 20, Perkin Elmer, Waltham, MA, USA). The initial optical density of 1.0 mL in a disposable plastic cuvette was recorded for 30 s, after which expansin (CsExp1) or snail powder extract was added to the suspension. The cuvette was then gently inverted to mix the solution and returned to the spectrophotometer, and the optical density was recorded for a further 120 s. The effects of horseradish peroxidase (Sigma-Aldrich, P6140) and H_2_O_2_ on the optical density of suspensions from sunflower hypocotyls were also tested.

### 4.8. Statistical and Mathematical Analysis

Where error bars are given, experimental data are presented as mean ± standard deviation, calculated using Microsoft Office Excel (Microsoft Corporation, Redmond, WA, USA). Strain at time t relative to time 0 are true strains calculated from Ln(L_t_/L_0_), where L_t_ is the segment length at time t and L_0_ is the length at time 0 (time 0 was immediately before the stress applied to the segment was increased unless otherwise stated). Modeling and parameterization of extensiometric data used the Microsoft Office Excel solver to minimize the squares of differences between modeled and experimental data.

## Figures and Tables

**Figure 1 plants-10-01263-f001:**
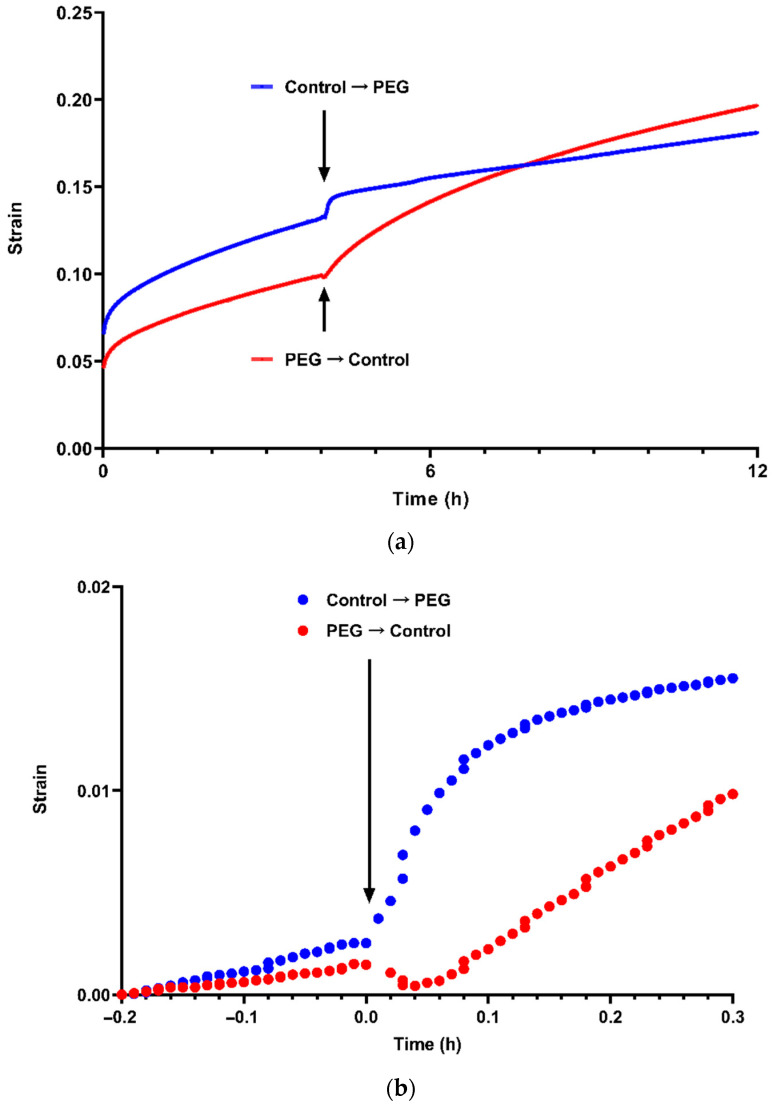
Effects of exchanging MES buffer for the same buffer containing PEG and the opposite exchanges, altering the osmotic potential from 0 MPa to −0.62 MPa or from −0.62 MPa to 0 MPa (**a**) and (**b**) on extension of cell walls from sunflower hypocotyls, (**c**) on bacterial cellulose and (**d**) on bacterial cellulose composite with pectin and xyloglucan. Arrows indicate the times at which osmotic potential was increased or reduced. Strain = Ln(L_t_/L_0_), where L_t_ = length at time t and L_0_ = length at plot time zero (these were lengths immediately before load was increased in plots (**a**), (**c**) and (**d**) and lengths 12 min before buffers were exchanged in (**b**). (**a**) Sunflower hypocotyls extending under an applied load of 0.196 N (applied at 0 h). Blue: 0 MPa → −0.62 MPa; Red: −0.62 MPa → 0 MPa. (**b**) Detail of data in Figure 1a illustrating changes in strain immediately after increasing or decreasing water potential. Blue: 0 MPa → −0.62 MPa; Red: −0.62 MPa → 0 MPa. (**c**) Strips of bacterial cellulose extending under an applied load of 0.196 N (applied at 0 h). Blue: −0.62 MPa → 0 MPa → −0.62 MPa; Red: 0 MPa → −0.62 MPa → 0 MPa. (**d**) Strips of bacterial cellulose composite with pectin and xyloglucan extending under an applied load of 0.098 N (applied at 0 h). Blue: −0.62 MPa → 0 MPa → −0.62 MPa; Red: 0 MPa → −0.62 MPa → 0 MPa.

**Figure 2 plants-10-01263-f002:**
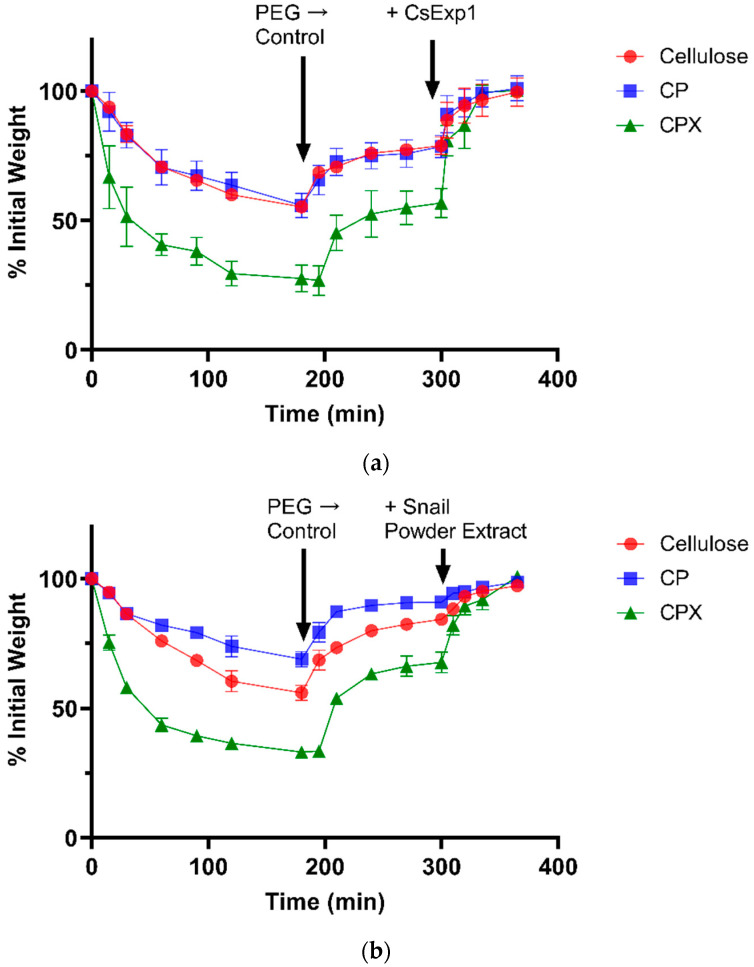
Time course of changes in hydration in pieces of bacterial cellulose (red circles), bacterial cellulose composite with apple pectin (blue squares), and bacterial cellulose composite with apple pectin and xyloglucan (green triangles) after the materials were immersed in buffer with an osmotic potential of −0.62 MPa (at 0 h), then returned to control buffer (first arrow), and finally treated with 11.2 μg of α-expansin (second arrow in (**a**)) or 200 μL of snail powder extract (second arrow in (**b**)). Error bars indicate S.D. (*n* = 3).

**Figure 3 plants-10-01263-f003:**
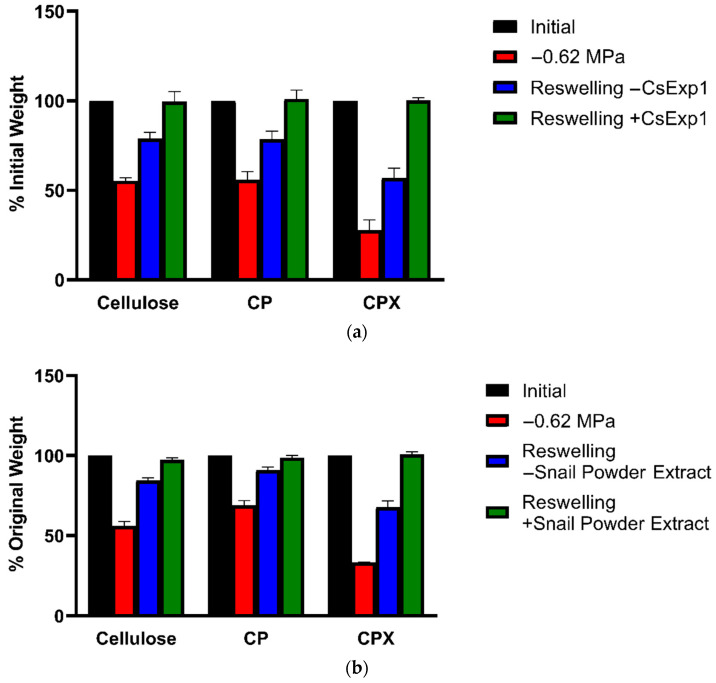
Summary of changes in hydration in pieces of bacterial cellulose, bacterial cellulose composite with apple pectin (CP), and bacterial cellulose composite with apple pectin and xyloglucan (CPX) as a percentage of initial weight (black fill), after 180 min at −0.62 MPa (red fill), after 120 min in control buffer (blue fill), and after treatment with 11.2 μg of α-expansin (green fill in (**a**)) or 200 μL of snail powder extract (green fill in (**b**)). Error bars indicate S.D. (*n* = 3).

**Figure 4 plants-10-01263-f004:**
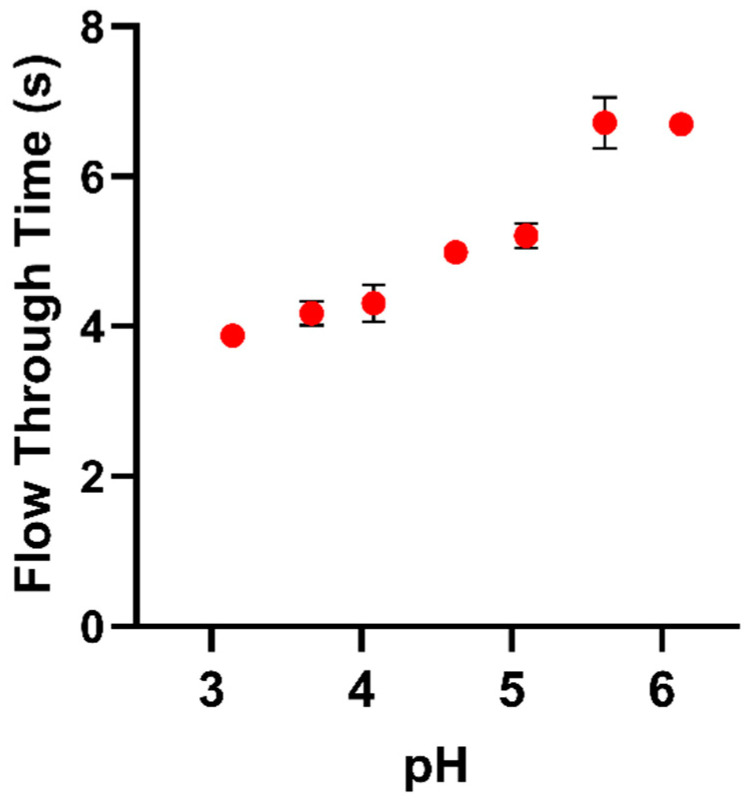
The effect of pH on time taken for 2% (*w*/*v*) solutions of apple pectin of pH 3.14, 3.67, 4.08, 4.63, 5.1, 5.62, and 6.13 containing 5 mM CaCl_2_ to flow between two graduations of a 1 mL glass pipette. Error bars indicate S.D. (*n* = 3).

**Figure 5 plants-10-01263-f005:**
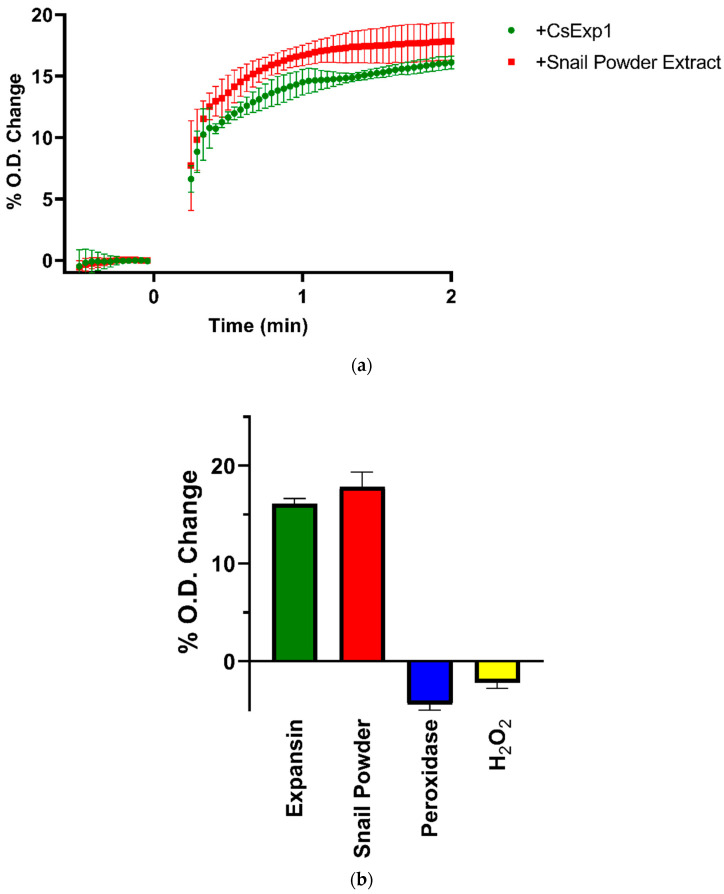
Effects of addition of 14 µg CsExp1 expansin or 200 µL snail powder extract on swelling of cell wall fragments from sunflower hypocotyls in suspension, measured by their effect on proportional O.D. at λ = 750 nm relative to initial O.D. (**a**) Time course of change in O.D. after addition at 0 s of expansin (green circles) or 200 µL snail powder extract (red squares). (**b**) Summary of O.D. change 2 min after addition of expansin (green fill), snail powder extract (red fill), 2 units of horseradish peroxidase (blue fill), and H_2_O_2_ to give a final concentration of 84 mM H_2_O_2_ (yellow fill). Error bars indicate S.D. (*n* = 3).

**Figure 6 plants-10-01263-f006:**
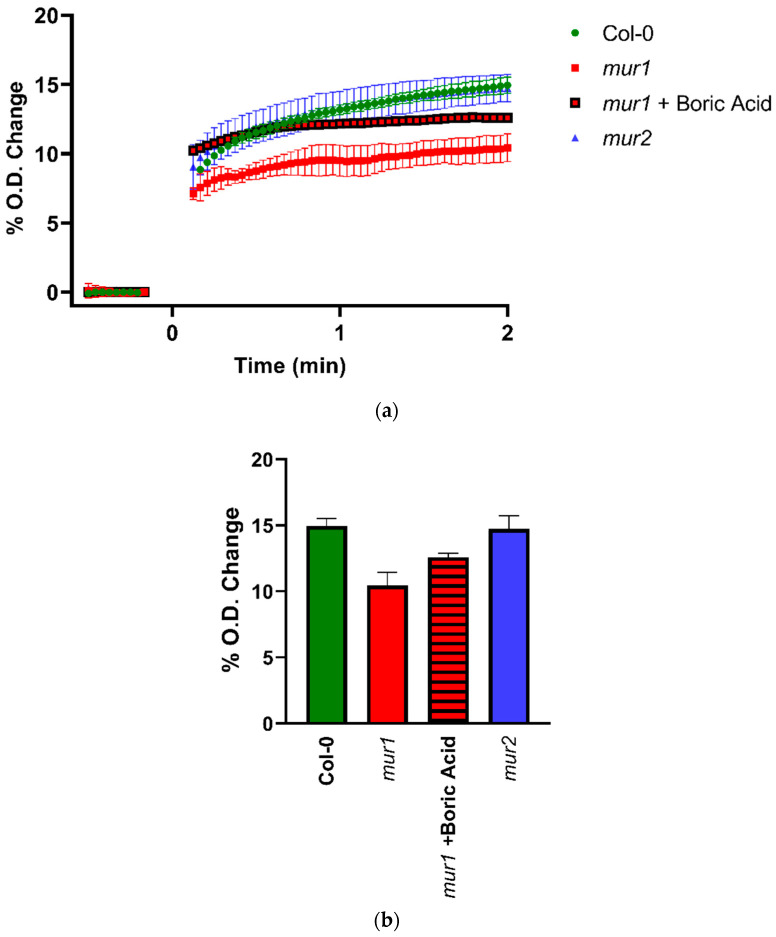
Effects of addition of 14 µg CsExp1 expansin on swelling of cell wall fragments from Col-0, *mur1*, and *mur2* Arabidopsis leaves and stems suspended in buffer, measured by its effect on proportional O.D. at λ = 750 nm relative to initial O.D. (**a**) Time course after addition of expansin at 0 s to Arabidopsis cell wall fragments from wild-type seedlings (green circles), *mur1-2* seedlings (without boric acid pretreatment, red squares; with boric acid pretreatment, red squares with black border) and *mur2* seedlings (blue triangles). (**b**) Summary of the O.D. increase 2 min after addition of expansin to Arabidopsis cell wall fragments from wild-type seedlings (green fill), *mur1-2* seedlings (red fill), fragments from *mur1-2* seedlings pretreated with boric acid (red fill with hatching), and fragments from *mur2* (blue fill). Error bars indicate S.D. (*n* = 3).

## Data Availability

Data is contained within the article.

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
