# Peer review of "Plant Cell Wall Hydration and Plant Physiology: An Exploration of the Consequences of Direct Effects of Water Deficit on the Plant Cell Wall"

_plants, 2021, doi:10.3390/plants10071263_

Round 1

Reviewer 1 Report

I found your research appropriate for publication in Plants. However, in my opinion, the article is descriptive and some changes in the text should be performed before the publication (comments listed below).

Comments:

  1. In my opinion, the introduction is too long and should be shortened. Also, some sentences are repeated several times in various words.
  2. In all fig-s please take a, b, etc. letters out of the graphs and put them from the left side in bold and bigger -it is difficult to look for such tiny letters that are closed to the scale.
  3. Fig 1. Legend contains repeats also, like: “of 0.196 N (applied at 0 h) were initially immersed in control 238 buffer which was exchanged for buffer with an osmotic potential of -0.62 MPa at the first arrow 239 and then returned to control buffer at the second arrow (red)” or information about arrows. Please try to put it at the end of the legend for a,b,c,d together. The difference is only in d - 098N. in fig itself, please add what is blue and red -for ex.: Red-cont/ blue—0.62Mpa. Please add an explanation for strain calculation in legend or results in the description of fig 1.
  4. Lines 169-170. Please transfer the description of mes buffer to methods, if needed at all.
  5. Line 200-“when the osmotic potential was decreased and when it was decreased. Is the second decrease should be “increase”?
  6. Line 213-References should be in numbers (Thomson 2001).
  7. Fig 2. Please add sign explanation within graphs 2 a and c, like blue square-CP and etc. You also have repeats in the legend of fig 2. Please write it once for several of them.
  8. Lines 302-303-Please transfer to the methods, leave only the final concentration.
  9. Line 317-Please remove 1 before SD (The same in fig 4). Also please write s.d. in uppercase
  10. Fig 3 legend: repeats, can you combine and write the same thing only once?
  11. Fig 4 A) legend, a and b: “λ = 750 nm” “after addition of 14 μg CsExp1” and etc.. Why do you need to repeat the same things in the one fig?

B)Can you add mur1boric treatment also into 4a?

  1. Please transfer fig 5 to supplement, or results or organize results and discussion together (results and discussion format is acceptable in Plants). In my opinion, fig5 b should be in a supplement.
  2. Please shorten the discussion, especially the salt stress part-you did not perform exp-s under salinity. Salt stress is a complex one that includes several primary and secondary stresses, that also will affect cell walls in their way. Your discussion is highly descriptive, please stay focused.
  3. Please transfer conclusions at the end of the discussion part and shorten them significantly.

Best regards.

Author Response

I found your research appropriate for publication in Plants. However, in my opinion, the article is descriptive and some changes in the text should be performed before the publication (comments listed below).

Thanks for your thoughtful and helpful comments. I do not disagree with anything that you have said and I believe that the changes that you have suggested make for a considerably improved submission. I had intended that this manuscript include theoretical and review elements in addition to the data, which is why I chose “Communication” as a description. However, I take your point that even so it is unfocussed in places and have been pleased to re-edit in the light of your comments.

 Comments:

  1. In my opinion, the introduction is too long and should be shortened. Also, some sentences are repeated several times in various words.

I have tried to streamline the text and removed the initial paragraph and that on tree height, as well as some other edits. As a result the Introduction has been reduced from 130 lines to 100 lines.

  1. In all fig-s please take a, b, etc. letters out of the graphs and put them from the left side in bold and bigger -it is difficult to look for such tiny letters that are closed to the scale.

This has been done.

  1. Fig 1. Legend contains repeats also, like: “of 0.196 N (applied at 0 h) were initially immersed in control 238 buffer which was exchanged for buffer with an osmotic potential of -0.62 MPa at the first arrow 239 and then returned to control buffer at the second arrow (red)” or information about arrows. Please try to put it at the end of the legend for a,b,c,d together. The difference is only in d - 098N. in fig itself, please add what is blue and red -for ex.: Red-cont/ blue—0.62Mpa. Please add an explanation for strain calculation in legend or results in the description of fig 1.

To be honest I had the legends cumbersome as well. I have tried to streamline all of them to reduce repetition and improve clarity.

  1. Lines 169-170. Please transfer the description of mes buffer to methods, if needed at all.

This has been done.

  1. Line 200-“when the osmotic potential was decreased and when it was decreased”. Is the second decrease should be “increase”?

Yes, this has been corrected.

  1. Line 213-References should be in numbers (Thomson 2001)

This has been corrected (I used Harvard referencing in earlier drafts and this one slipped through when I changed to numerical).

  1. Fig 2. Please add sign explanation within graphs 2 a and c, like blue square-CP and etc. You also have repeats in the legend of fig 2. Please write it once for several of them.

This has been done.

  1. Lines 302-303-Please transfer to the methods, leave only the final concentration.

This has been done.

  1. Line 317-Please remove 1 before SD (The same in fig 4). Also please write s.d. in uppercase.

This has been done.

  1. Fig 3 legend: repeats, can you combine and write the same thing only once?

This has been done.

  1. Fig 4 A) legend, a and b: “λ = 750 nm” “after addition of 14 μg CsExp1” and etc.. Why do you need to repeat the same things in the one fig?

This has been changed as you suggest.

B)Can you add mur1boric treatment also into 4a?

This has been done (in what is now Fig. 6a).

  1. Please transfer fig 5 to supplement, or results or organize results and discussion together (results and discussion format is acceptable in Plants). In my opinion, fig5 b should be in a supplement.

Fig. 5a has been moved to where the effect of pectins on hydration appears in the Results. As a result of this and the separation of Fig. 2 into two separate figures at the suggestion of one of the other referees  Fig. 5a is now Fig. 4. Fig. 5b in the Appendix as Fig. S2. As a result of these changes, Figs 2a and 2c become Figs 2a and 2b and Figs 2b and 2d become Figs 3a and 3b, Fig. 3 is now Fig. 5 and Fig. 4 is now Fig. 6.

  1. Please shorten the discussion, especially the salt stress part-you did not perform exp-s under salinity. Salt stress is a complex one that includes several primary and secondary stresses, that also will affect cell walls in their way. Your discussion is highly descriptive, please stay focused.

I broadly agree about the salt stress section, which I had substantially included as a way of framing the discussion of the mur-1 data presented as while I can find no publications about mur-1 and water stress, there are a number about salt stress (and similar observations in some of the other mutants with altered RGII cross linking). I have reframed this section around RGII, which I hope is more appropriate for the data presented. The section has been reduced to 154 lines from 184 lines, which I hope will be acceptable give the theme of the special issue.

  1. Please transfer conclusions at the end of the discussion part and shorten them significantly.

This has been done.

Best regards.

Reviewer 2 Report

In general, the article presents some very interesting and important results. However, it’s difficult to read and understand.

For me, it would be interesting to compare homogalacturonans with and without methyl-esterification under water stress. The literature on this topic is inconclusive. Of course, it’s only a general remark, I do not expect the authors to perform more experiments.

Introduction

For someone not from the field, the introduction was both too long, too unfocused and difficult to read. For instance, I did not know what’s Donnan effect.

I would remove the first paragraph. It’s not contributing much to the article.

L53-57 “There is good reason to expect that changes in the water content of plant cell walls  could affect their mechanical behaviour directly. In polymer matrices the free volume between polymer molecules is thought to have substantial effects on their properties and  behaviour. Increasing temperature makes such materials easier to deform and causes transitions from more solid (glassy) to softer (viscoelastic, rubbery and flow) behaviours because it causes an increase in spacing between macromolecules [8].” – Temperature effects have no relevance to this study, the intro is too long and confusing as is.

L62 – “Certainly, the degree of hydration alters the strength of  pectin films several fold [10].” – alters how? To which direction?

L76-79 “Changes in wall hydration, spacing and tissue biomechanics as a result of plant water  status would be expected to have a direct impact upon plant growth, and additionally reduced spacing might impede movement of enzymes and other molecules within the  wall [14], and indeed reduce tissue hydraulic conductivity and amplify water potential  gradients as a result (e.g. [15]).” Maybe I misunderstood, but how exactly cell wall hydration status would change the water potential inside the cell? The cell wall is water permeable in most cells.

L93-96 “The quantity of water bound per monosaccharide unit of chitosan, alginate and cellulose in paper differ, as does the way that this relationship changes with water activity [16], and interactions between Arabidopsis cell walls and water were modified as their  wall polysaccharides were sequentially extracted and so these factors can clearly alter the  relationship between water potential and degree of hydration [17].” – please fix this sentence. What paper are you referring to in the beginning and then suddenly skipping to Arabidopsis?

L89-104 - Please explain what polymers and side chains you mean. I guess you are talking about pectins, but it’s really unclear from the text. The introduction is unclear.

“An additional instance where plant tissues can experience constant low or very low 135 water potentials are the cell walls of tissues at the top of trees, which have been estimated 136 to reach ≤ -2 MPa [25], conditions comparable to extreme drought, because of the weight 137 of the water column in the xylem. Leaves at the top of tall redwood trees were much 138 smaller than nearer to the ground, but from a substantial increase in the leaf dry mass per 139 unit area it seems that the quantity of cell wall material may have increased with height 140 [25]. This does not necessarily reflect differences in wall structure or composition, for ex-141 ample it may simply be that the cells were smaller because turgor pressures decreased 142 with height and so their area to volume ratios were higher, but it is nonetheless an intri-143 guing observation not least because it has been argued that while the turgor pressures in 144 leaves near to the top of the trees were lower than in those near to the ground, they did 145 not appear to be particularly extreme, presumably because of osmotic adjustment [26]. It 146 may therefore be worth exploring whether there are adaptations in the walls of these tis-147 sues that allow them to function at such extreme water deficits.” – This paragraph should be removed, it’s irrelevant to this study (though indeed could be a nice topic for another article.

Results

“Observing whether polyethylene glycols (PEGs) of a range of molecular weights  caused plasmolysis (exerting osmotic pressure at the cell membrane) or the cell collapse  called cytorrhysis was used to determine size of pores plant cell walls [27], but these experiments also demonstrated that PEG with a molecular weight of > 4 kDa can be used to  apply osmotic potentials to plant cell walls. It has previously been reported that such  treatments reduce the extension rate of plant cell wall material in constant-load extensiometer (also known as creep) measurements [11, 12, 24].” -  unclear paragraph. Please rephrase. Why did you use PEG of different molecular weight? The paragraph does not explain that sufficiently. What was the objective of your experiment?

Please explain what’s Poisson’s ratio and what are Kelvin Elements. Those are engineering terms, and should be explained to the readership of this botanical journal.

Fig. 1- Vey unclear. First of all, please add A,B,C,D to the panels. Second, write near each arrow what was added. Add the legend inside one of the panels explaining what is red and what is blue. The figure is difficult to read and understand.

Fig. 2 – again, the figure was not edited properly. The panels are not aligned, and lack the letters (ABCD). More changes needed- similar as Fig. 1. I would split this figure into two- one with panels A and C, the second with B and D.

I suggest explaining a bit about the snail powder in the Results section. It’s an interesting method.

M&M

How many repeats were in each experiment?

Author Response

In general, the article presents some very interesting and important results. However, it’s difficult to read and understand.

For me, it would be interesting to compare homogalacturonans with and without methyl-esterification under water stress. The literature on this topic is inconclusive. Of course, it’s only a general remark, I do not expect the authors to perform more experiments.

Thank you for your thoughtful and helpful comments. On reflection, I feel that the writing was not as clear as it should have been and I have re-edited to try to make the text simpler and more direct throughout. Yes, the issue of methylesterification is a really interesting one and something we would like to return to.

Introduction

For someone not from the field, the introduction was both too long, too unfocused and difficult to read. For instance, I did not know what’s Donnan effect.

I have tried to streamline the text throughout the manuscript and as a result, together with removal of paragraphs as you suggest the Introduction has been reduced from 130 lines to 100 lines.

I have included more explanation of terms such as Donnan effect, which plant scientists may not be familiar with.

I would remove the first paragraph. It’s not contributing much to the article.

I had included this because of the theme of the special issue but I have removed it as suggested.

L53-57 “There is good reason to expect that changes in the water content of plant cell walls  could affect their mechanical behaviour directly. In polymer matrices the free volume between polymer molecules is thought to have substantial effects on their properties and  behaviour. Increasing temperature makes such materials easier to deform and causes transitions from more solid (glassy) to softer (viscoelastic, rubbery and flow) behaviours because it causes an increase in spacing between macromolecules [8].” – Temperature effects have no relevance to this study, the intro is too long and confusing as is.

There are interesting and important connections between temperature transitions and physical behaviour, but on reflection I see that I either needed to explain them fully or leave them out. I have removed this passage.

L62 – “Certainly, the degree of hydration alters the strength of  pectin films several fold [10].” – alters how? To which direction?

This line has been edited to make it clearer that the films become stiffer as their hydration decreases.

L76-79 “Changes in wall hydration, spacing and tissue biomechanics as a result of plant water  status would be expected to have a direct impact upon plant growth, and additionally reduced spacing might impede movement of enzymes and other molecules within the  wall [14], and indeed reduce tissue hydraulic conductivity and amplify water potential  gradients as a result (e.g. [15]).” Maybe I misunderstood, but how exactly cell wall hydration status would change the water potential inside the cell? The cell wall is water permeable in most cells.

What I had in mind was the additive effect of cell wall resistance to water movement across tissues, which some authors do claim can amplify water potential gradients (much as narrowing a stream would). I have tried to clarify this.

L93-96 “The quantity of water bound per monosaccharide unit of chitosan, alginate and cellulose in paper differ, as does the way that this relationship changes with water activity [16], and interactions between Arabidopsis cell walls and water were modified as their  wall polysaccharides were sequentially extracted and so these factors can clearly alter the  relationship between water potential and degree of hydration [17].” – please fix this sentence. What paper are you referring to in the beginning and then suddenly skipping to Arabidopsis?

I have removed mention of the Arabidopsis paper here and tried to tidy up the phrasing of the section.

L89-104 - Please explain what polymers and side chains you mean. I guess you are talking about pectins, but it’s really unclear from the text. The introduction is unclear.

I was referring to general principles of interaction between polymers and water, but you are correct that I would expect that most pectins would be the main cause of these effects in plant cell walls. I have rephrased to clarify that the sentence refers to theoretical principles and added a second making specific reference to pectins.

“An additional instance where plant tissues can experience constant low or very low 135 water potentials are the cell walls of tissues at the top of trees, which have been estimated 136 to reach ≤ -2 MPa [25], conditions comparable to extreme drought, because of the weight 137 of the water column in the xylem. Leaves at the top of tall redwood trees were much 138 smaller than nearer to the ground, but from a substantial increase in the leaf dry mass per 139 unit area it seems that the quantity of cell wall material may have increased with height 140 [25]. This does not necessarily reflect differences in wall structure or composition, for ex-141 ample it may simply be that the cells were smaller because turgor pressures decreased 142 with height and so their area to volume ratios were higher, but it is nonetheless an intri-143 guing observation not least because it has been argued that while the turgor pressures in 144 leaves near to the top of the trees were lower than in those near to the ground, they did 145 not appear to be particularly extreme, presumably because of osmotic adjustment [26]. It 146 may therefore be worth exploring whether there are adaptations in the walls of these tis-147 sues that allow them to function at such extreme water deficits.” – This paragraph should be removed, it’s irrelevant to this study (though indeed could be a nice topic for another article.

 This section has been removed.

Results

“Observing whether polyethylene glycols (PEGs) of a range of molecular weights  caused plasmolysis (exerting osmotic pressure at the cell membrane) or the cell collapse  called cytorrhysis was used to determine size of pores plant cell walls [27], but these experiments also demonstrated that PEG with a molecular weight of > 4 kDa can be used to  apply osmotic potentials to plant cell walls. It has previously been reported that such  treatments reduce the extension rate of plant cell wall material in constant-load extensiometer (also known as creep) measurements [11, 12, 24].” -  unclear paragraph. Please rephrase. Why did you use PEG of different molecular weight? The paragraph does not explain that sufficiently. What was the objective of your experiment?

I have tried to clarify this passage. I was referring to the experiments on cell wall pore size by Carpita et al., which suggested use of large PEGs to modulate water content in walls to me. I have edited the paragraph to try to make it clearer and expanded on the principles by which PEG has this effect.

Please explain what’s Poisson’s ratio and what are Kelvin Elements. Those are engineering terms, and should be explained to the readership of this botanical journal.

This has been done. The mathematical relationships have also been added as footnotes.

Fig. 1- Vey unclear. First of all, please add A,B,C,D to the panels. Second, write near each arrow what was added. Add the legend inside one of the panels explaining what is red and what is blue. The figure is difficult to read and understand.

The letters were present in the panels but not clearly enough. At the suggestion of another referee they have been taken out of the panels and added next to the figures. Legends have been added to the panels adjacent to the arrows, which I hope will make the treatments clearer. The figure legends have been edited to make them more streamlined throughout.

Fig. 2 – again, the figure was not edited properly. The panels are not aligned, and lack the letters (ABCD). More changes needed- similar as Fig. 1. I would split this figure into two- one with panels A and C, the second with B and D.

I have made these changes.

I suggest explaining a bit about the snail powder in the Results section. It’s an interesting method.

This has been done. Unfortunately this product has been discontinued by Sigma, but some of their crude snail beta-glucuronidases appear to retain some expansin activity.

M&M

How many repeats were in each experiment?

“n = 3” was included in many of the figure legends. The creep extensiometry traces rarely map well enough for data to be pooled and so examples are presented but I have included a statement that each experiment was run twice, with duplicates in each experiment.